# A Virtual Inner Ear Model Selects Ramped Pulse Shapes for Vestibular Afferent Stimulation

**DOI:** 10.3390/bioengineering10121436

**Published:** 2023-12-18

**Authors:** Joseph Chen, Jayden Sprigg, Nicholas Castle, Cayman Matson, Abderrahmane Hedjoudje, Chenkai Dai

**Affiliations:** 1Aerospace and Mechanical Engineering, University of Oklahoma, Norman, OK 73019, USA; 2School of Medicine and Biomedical Engineering, University of Oklahoma, Norman, OK 73019, USA; 3Biomedical Engineering Department, Johns Hopkins University, Baltimore, MD 21218, USA

**Keywords:** bilateral vestibular deficiency, vestibular prosthesis, electrical pulse, ramped pulse

## Abstract

Bilateral vestibular deficiency (BVD) results in chronic dizziness, blurry vision when moving the head, and postural instability. Vestibular prostheses (VPs) show promise as a treatment, but the VP-restored vestibulo-ocular reflex (VOR) gain in human trials falls short of expectations. We hypothesize that the slope of the rising ramp in stimulation pulses plays an important role in the recruitment of vestibular afferent units. To test this hypothesis, we utilized customized programming to generate ramped pulses with different slopes, testing their efficacy in inducing electrically evoked compound action potentials (eCAPs) and current spread via bench tests and simulations in a virtual inner model created in this study. The results confirmed that the slope of the ramping pulses influenced the recruitment of vestibular afferent units. Subsequently, an optimized stimulation pulse train was identified using model simulations, exhibiting improved modulation of vestibular afferent activity. This optimized slope not only reduced the excitation spread within the semicircular canals (SCCs) but also expanded the neural dynamic range. While the model simulations exhibited promising results, in vitro and in vivo experiments are warranted to validate the findings of this study in future investigations.

## 1. Introduction

Bilateral vestibular deficiency (BVD) is a disabling condition characterized by chronic dizziness, blurry vision during head movement, and postural instability. This condition presents significant challenges to individuals, and for those who have not responded to vestibular rehabilitation, treatment options remain limited. To address this medical challenge, recent advancements have led to the development of vestibular prostheses (VPs). In recent years, VPs have been developed [1,2,3,4] with the capability to detect three-dimensional (3D) head rotation and restore a 3D vestibulo-ocular reflex (VOR) in animals affected by BVD caused by gentamicin ototoxicity. Encouraging results have been observed in chinchillas and rhesus monkeys [5,6,7], demonstrating the potential of VPs.

However, the VOR gain achieved in human clinical trials [8] has fallen short of expectations, highlighting the need for further improvements in the performance of VPs. A potential contributing factor to the suboptimal VOR gain observed in VP clinical trials may stem from the stimulation protocol’s suitability, particularly concerning the predominantly elderly subjects involved. Although some patients may still experience reasonable improvements in balance function with a low VOR gain, a higher VOR gain has the potential to enhance the recovery of vestibular function facilitated by vestibular implants. Currently, the VP utilizes an electrical pulse composed of two phases of opposite polarity to ensure charge-balanced stimulation, and it adopts a rectangular shape. While this configuration has demonstrated safety in both animals and humans, it may not represent the most efficient means of stimulating neurons.

The utilization of ramped pulses for vestibular afferent stimulation offers several potential advantages, including reducing channel interaction and expanding the perceptual dynamic range. Ramped pulses have the capability to selectively activate specific vestibular nerve fibers, thereby increasing the number of independent channels for information transmission and mitigating channel interaction. In terms of the perceptual dynamic range, individuals with VPs often experience a limited electrical dynamic range of 0 to 400 firings per second (fps) in contrast with the wider range observed in individuals with normal vestibular function. This limited range is partially attributed to the synchronous neural activity generated by electric stimulation across the stimulated neural population. However, in natural stimulation, there is less synchronization across fibers and more variability within fibers due to the stochastic properties of the hair cell vestibular ganglion neuron (VGN) synapse. By incorporating ramped pulses, the expression of ion channels can vary, reintroducing a certain level of stochasticity and enabling graded sensitivity to ramped pulses across VGNs. This gradual recruitment of vestibular nerve fibers as the current level mirrors the natural balance and expands the dynamic range. Furthermore, studies [9,10] have shown that ramped pulses are more power-efficient than rectangular pulses in models of the auditory nerve, and recent experiments conducted by Yip et al. have demonstrated the efficiency of novel non-rectangular pulse shapes in human CI experiments, and their previous studies suggest several advantages of employing ramped pulses over rectangular pulses for CI stimulation.

Building upon previous studies [9,10,11,12,13,14,15,16,17] conducted on cochlear implants, it is logical to propose that the slope of the ramp-up/down pulses may influence the recruitment of vestibular afferent units during electrical stimulation, ultimately leading to improvements in VOR gain. To explore this hypothesis, a computational model was developed, allowing for the manipulation of the slopes of the ramped pulses, aiming to enhance VOR gain via simulations. The primary objective of this study is to contribute to the advancement of vestibular prostheses that can effectively restore stable vision and balance in individuals suffering from chronic disabilities. The anticipated outcomes of this research are expected to provide valuable insights into enhancing VOR gain, especially regarding senior patients undergoing vestibular prosthesis interventions.

## 2. Materials and Methods

### 2.1. Overall Approach

We adopted a standard pulse train as our current stimulation protocol for animal studies and human clinical trials [8]. To enhance the protocol, we customized the ramp-up and ramp-down slopes using programming developed in our laboratory. Subsequently, we evaluated the effectiveness of various edited stimulation protocols with different ramp-up and ramp-down slopes using computational modeling simulations and bench tests in a 3D-printed ear. Our criterion for selecting the optimal parameters was based on the onset of electrically evoked compound action potentials (eCAPs) and the alignment of the electrically evoked vestibulo-ocular reflex (eeVOR).

### 2.2. Stimuli Editing

Stimuli were created using our laboratory’s custom-made software program and transmitted to the implanted electrode array via a stimulator platform comprising an AM stimulator and Keithley pulse generator (Keithley 2601B, Tektronix, Beaverton, OR, USA). Digital signal processing was employed using a Tucker Davis Technologies digital signal processor (RZ6, TDT, Alachua, FL, USA) to generate and record trigger pulses for each stimulus. Our testing encompassed eight waveforms, comprising four pulse shapes and two polarities (Figure 1).

The rectangular pulse shape (Rec) featured a square waveform with a fixed phase–amplitude. The ramped shapes were defined based on their slope, representing the rate at which the injected current linearly increased or decreased over time. In ramp-up pulses, both the first and second phases exhibited a ramped slope from zero at the phase onset to a specified amplitude current level at the phase offset. In ramp-down, both phases exhibited a slope that ramped from a specified amplitude current level at the phase onset to zero at the phase offset.

Both anodic-first and cathodic-first polarities were tested. Notably, it is important to mention that the ramped current slope was approximated with a current step size of 20 μA due to a hardware limitation with our pulse generator. The temporal resolution of the pulse generator was 2 μs. For rectangular pulses, the charge per phase was calculated as the pulse width multiplied by the current level amplitude. For ramped pulses, the charge per phase was calculated as the pulse width multiplied by the rectangular part of the pulse plus the triangular (ramp-up) current level amplitude divided by two.

### 2.3. Computational Modeling

To ascertain the ideal ramp slope for the new stimulation protocol, we employed a virtual model of the rhesus monkey ear to perform simulations. The model’s specific characteristics were previously outlined in our studies [18,19,20]. Our model design adopted a methodology akin to cochlear implant stimulation models, comprising two primary components [20].

Firstly, we employed a finite element volume conductor model to estimate the current densities along the ampullary nerves. This model considers electrode arrangements and applied currents, enabling accurate estimations. Secondly, we utilized a spike initiator model to simulate the transduction of current density along each nerve branch into action potentials in prototypical vestibular afferents. This component accurately captures the neural response to the applied electrical stimulation.

The CT and MRI data underwent a co-registration and segmentation process facilitated by Amira 3D visualization software. This crucial procedure enabled the precise identification of individual ampullary and macular nerves, as well as the principal trunk of the vestibular nerve, facial nerve, cristae, and maculae at a notably high resolution. Subsequently, the segmented slice data were seamlessly incorporated into 3D surface models, and Amira was employed to generate tetrahedral meshes. The resulting models were meticulously crafted and yielded accurate depictions of the neural structures, along with the perilymphatic spaces of the vestibule and semicircular canals (Figure 2).

The presence of metal electrodes introduces distortions that hinder the accurate depiction of nerve anatomy solely via MRI data for implanted specimens. To address this challenge, we pursued an alternative approach. By conducting a micro-CT scan of the implanted labyrinth, we achieved an approximate co-registration with a normative labyrinth model that already included nerves segmented from micro-MRI data. This allowed us to customize the model for each implanted animal, incorporating individualized electrode positions, ampulla shapes, and semicircular canal orientations. The microelectrodes in the implanted animal were virtually positioned in equivalent locations within the normative model. To enhance the resolution around the electrodes, the normative model underwent re-meshing.

For the prediction of relative current intensities and gradients along the central axis of each ampullary nerve, assuming quasistatic conditions, we utilized finite element solvers in COMSOL Multiphysics (COMSOL Inc., Burlington, MA, USA). Additionally, the orientation of the semicircular canal axis relative to the skull in each implanted animal could be directly measured from their CT scans. This valuable information was utilized to determine the corresponding axis of rotation for the prosthetically evoked eye velocity.

In our model, the ampullary nerves were considered to have homogeneous but anisotropic conductivity. The effective stimulus that connected the finite element and neurophysiologic components of the model was the current density field along the central axis of the distal portion of each nerve. To capture the temporal dynamics of action potential initiation, we incorporated spike initiation models [21,22] and repolarization models [23,24] into our simulations.

In the initial phase, we employed a straightforward model, taking into consideration that channel interaction and electrode selectivity mainly depend on the relative current density or current density gradient along the nerve branches, irrespective of the particular spike initiator used. Our primary emphasis was on capturing the collective sensitivity of vestibular primary afferent fibers in each ampullary nerve to a specific current intensity. Consequently, at this stage, we temporarily set aside the complexities associated with action potential generation to focus on the broader response patterns.

The model computed the predicted eye velocity axes resulting from the activation of each electrode pair by summing the semicircular canal axes, weighted with the relative magnitude of the axial current along the corresponding ampullary nerves. These predictions were directly compared with the observed eye velocity axes measured during the experiments conducted in previous studies [5,6]. The model underwent iterative refinement based on the level of disparity between the model predictions and the measurements. Subsequent models incorporated more complex aspects of spike initiation dynamics as we progressed in our research.

### 2.4. Simulation of eCAPs

To calculate the electrically evoked compound action potentials (ECAPs) recorded via inactive electrodes in the implanted lead, we employed a computational model of vestibular afferent stimulation (VAS). The computational model comprised two primary components: (1) a finite element model (FEM) of a VAS lead implanted in the three semicircular canals (SCCs), and (2) multicompartment cable models of vestibular neuron axons. The procedure for calculating eCAPs during VAS involved the following steps: (1) Using the FEM, we computed the extracellular voltages generated in the vestibular neurons and the surrounding tissues during VAS. (2) We evaluated the direct axonal response to VAS by applying the extracellular voltages to the sensory axon models, obtaining transmembrane currents in response to the stimulation. (3) By utilizing a reciprocal FEM solution, we determined the voltage generated at each recording electrode, allowing us to calculate the VAS-induced eCAP.

Using the theorem of reciprocity, our computational framework allowed us to simulate and analyze the eCAPs resulting from vestibular afferent stimulation with the implanted lead. To simulate eCAP recordings, we applied the principle of reciprocity to calculate time-dependent voltages generated at each electrode via the axonal response to VAS.

In our approach, we applied a unit current source at each recording electrode, grounded the outer boundaries of the general thorax, and implemented Robin boundary conditions at the other electrodes. By solving the Laplace equation, we obtained model tissue voltages. These voltages were then interpolated onto each axonal compartment, representing the voltage impressed onto the recording electrode via a unit current source placed at the spatial location of each compartment.

To ascertain the electrically evoked compound action potentials (eCAPs), we integrated the voltages generated at each recording electrode with the scaled transmembrane currents obtained from independent compartments. This iterative process was carried out for each recording electrode, and bipolar differential recordings were derived by utilizing a nearby electrode as the reference signal, subsequently subtracted from each recording electrode. For the experimental processing simulation, we applied a low-pass Butterworth filter with a cutoff frequency of 7.5 kHz to each signal to replicate real-world conditions.

To evaluate the characteristics of our model eCAP recordings, we utilized two thresholds: the model sensory threshold (ST) and the discomfort threshold (DT). The ST was defined as the stimulus amplitude that resulted in the activation of at least 10% of the DC fibers in the spinal cord. As for the DT, it was set at 1.4 times the ST, based on established findings from previous studies [25,26,27,28]. We quantified the eCAP amplitude by calculating the difference between the P and N peak amplitudes (Figure 3).

### 2.5. Bench Test for New Simulation Protocol

The bench test for the stimulation protocol involved a series of complex steps. First, the power supply was connected to an appropriate power source and configured to generate the desired electrical pulses. The parameters, such as pulse duration, pulse amplitude, and frequency, were set according to the specific requirements of the testing protocol. Next, the electrodes were attached to the bench power supply, ensuring they were properly connected and securely attached to the subjects (the 3D-printed inner ear based on the uCT of a rhesus monkey’s temporal bone). The electrodes were placed at predetermined locations on the 3D-printed rhesus monkey ear, including the vestibular and cochlea regions. The impedance of 3D-printed ear sensory tissues is similar to that of real tissues, and the middle chamber was fluid-filled with 0.9% NaCl to simulate the endolymph fluid. Consistent electrode placement across different measurements is crucial for accurate comparisons. Once the electrodes were in place, initial measurements of the current were taken at the stimulated electrode and any nearby electrodes where we wanted to assess the current spread. While the stimulation was ongoing, the current was simultaneously measured at the stimulated electrode and the nearby electrodes using a current meter or multimeter. After completing the measurements, the recorded data were analyzed to look for patterns or changes in the current readings across different electrodes. The average current spread was calculated to evaluate any variations based on the stimulation parameters. The experiments were repeated multiple times with different stimulation parameters or electrode placements to ensure the reliability of the results.

### 2.6. Statistical Analysis

To assess and compare the overall effects of response amplitudes and latencies, we utilized factor analyses employing a mixed model with random effects in a statistical software program JMP (version 16.1, SAS Institute, Cary, NC, USA). We visually inspected plots of residuals against fitted values to evaluate the assumptions of normality and homogeneity. To evaluate the VOR threshold, we conducted a paired *t*-test. Pearson correlation analysis was employed to assess the correlations between the aVOR and eVOR thresholds as well as the growth function slopes. A two-tailed *p*-value was used to determine statistical significance, and we applied a significance level of α = 0.05 to all statistical tests.

## 3. Results

### 3.1. Ramped Shapes Can Evoke eVORs and eCAPs

An eCAP is a compound potential evoked in response to vestibular afferent stimuli. To first test if ramped pulses could generate an eCAP, we presented trains of both rectangular and ramped pulses to the electrode (E0) and measured the eCAPs in the first group of simulation in the FE rhesus monkey model. The four pulse shapes produced highly similar eCAP wave patterns, as shown in Figure 3. The lowest-threshold-induced eCAP was a 30-degree ramped pulse stimulus at an approximately 30 uA level in comparison with the threshold of 40 uA with the rectangular standard pulse stimuli (Figure 4).

### 3.2. Effects of Ramping-Up/Ramping-down on Vestibular Afferents

Figure 5n–p shows the computational modeling simulation can predict the eVOR with ramped pulse stimulation in three SCCs. The ramped 30-degree pulse train resulted in a robotic eVOR, and this indicates that the ramped pulse was effective in terms of recruiting vestibular afferent units. The eVOR amplitudes in three different components were plotted against various ramped slopes (15, 30, 45, and 60) in Figure 6a. The dynamic range represented spanned from the threshold to the charge level just before a facial nerve response was triggered or until the response reached saturation. Notably, different types of growth functions were observed across the four ramped slopes, suggesting the potential to select the optimal ramped slope to achieve the maximum eVOR at the same stimulation level. A comparison of the model’s afferent activity predictions with four ramped-slope pulses showed that the relationship between each canal-axis-specific component of the model predicted the 3D VOR response velocity to the proportion of fibers active in the corresponding ampullary nerve. The comparison data are shown in Figure 7 and indicate stimuli with 30-degree ramp-up pulses had the maximum response with less misalignment.

Ramped shapes demonstrate steeper slopes in the growth function of electrically evoked compound action potentials (eCAPs) compared with rectangular shapes. The eCAP growth function reflects the evoked synchronous activity, which is determined by the recruitment speed and number of synchronized neurons. To investigate whether the pulse shape affects the slope of the eCAP growth function, we calculated the individual slopes for each pulse shape.

A steeper growth function also implies that the maximum response can be achieved with a lower amount of charge. In most simulation and bench tests, the upper limit of electric stimulation was determined by the onset of facial nerve responses. The findings reveal that ramped pulses allowed for the use of significantly higher current levels while requiring lower amounts of charge to avoid evoking a facial nerve response compared with the utilization of rectangular pulses. Therefore, stimulation with ramped pulses could be advantageous if the reduction in charge required to reach the most comfortable level exceeds that needed to excite the facial nerve. However, further insight into this aspect is needed via clinical studies.

## 4. Discussion

This study presents the first simulation data on VP stimulation with ramped pulse shapes. We found that less charge is needed with ramped pulse shapes than with rectangular pulse shapes to evoke an eVOR response of similar amplitude. The 30-degree ramp-up pulse shape over both phases was most efficient in terms of a lower threshold and less misalignment to produce a response. Cathodic-first ramp-up pulses had lower thresholds than anodic-first pulses, which was also demonstrated in traces with the same interphase gap. Interestingly, the ramped pulse stimulation paradigm also showed that both anodic and cathodic phases could evoke an eVOR response and eCAP. This finding is in accordance with previous studies in cochlear implant studies [15]. Finally, our results demonstrate that specific ramped slopes exhibited higher efficiency compared with rectangular shapes, characterized by reduced misalignment and a higher recruitment rate. These findings provide support for the hypothesis that ramped pulse shapes can offer benefits such as an increased afferent recruitment rate and decreased misalignment, ultimately leading to the potential enhancement of VOR gain in vestibular prostheses (VPs). However, the mechanism behind this finding remains unclear, and more physiological measurements need to be performed to explore the possible cause. In the future, a series of complex physiological studies (e.g., patch-clamp measurements) should be undertaken to investigate the underlying mechanism.

Chronic profound bilateral loss of vestibular sensation has significant consequences, causing disabilities and imposing substantial costs on individuals and society alike. Regardless of the cause, such as ototoxic drugs or other inner ear insults, bilateral vestibular deficiency (BVD) results in several debilitating effects. These include impaired vision during head movement due to the failure of the VOR, imbalance leading to an increased risk of falls due to the compromised reflexes responsible for maintaining a stable posture, cognitive distraction stemming from the conscious effort required to complete tasks that are typically automatic, disorientation caused by the disrupted perception of the self in relation to the surrounding movement, and psychological distress [28,29,30,31].

Individuals with moderate bilateral vestibular deficiency (BVD) often rely on rehabilitation exercise regimens that promote multisensory mechanisms to improve gaze and posture stability [30]. However, while alternative oculomotor systems can partially compensate for a deficient vestibulo-ocular reflex (VOR), their effectiveness is limited by the delayed visual processing required, leading to failure during rapid head movements [31]. Gaze stabilization in the absence of a functional VOR involves predictive mechanisms, motor efference, and the cervico-ocular reflex [32]. Unfortunately, these systems are unable to cope with the rapid head movements encountered in everyday activities, such as driving, resulting in their failure [21,30]. Individuals with BVD who are unable to compensate for their vestibular loss experience chronic oscillopsia (a perception of illusory movement in the visual world), disequilibrium, and postural instability [27,28,29,30].

The electric stimulation provided by a current vestibular prosthesis (VP) was not efficient in terms of VOR gain in clinical trials [11] in comparison with results from rhesus monkey experiments [5,6]. One underlying problem is the poor efficiency with which information from electric pulses is transformed into vestibular afferent responses. A novel stimulation paradigm using ramped pulse shapes has recently been proposed to remedy this inefficiency. The primary motivation is a better biophysical fit to vestibular ganglion neurons with ramped pulses compared with the rectangular pulses used in most contemporary VPs. In this study, we examined two hypotheses: firstly, that ramped pulses offer more efficient stimulation compared with rectangular pulses, and secondly, that a rising ramp is more efficient than a declining ramp. Rectangular, rising ramped, and declining ramped pulse shapes were compared in terms of charge efficiency and discriminability, and threshold variability in VP stimulation modeling predictions. The tasks included single-channel threshold detection, the discrimination of pulse shapes, and threshold measurements across the electrode array. The results show that a reduced charge, but increased peak current amplitudes, was required at threshold and most comfortable levels with ramped pulses relative to rectangular pulses. The present findings show some benefits in charge efficiency with ramped pulses relative to rectangular pulses, that the direction of the ramped slope is of less importance, and that most simulations and bench tests could not reveal a significant difference between pulse shapes. The potential benefit of ramped pulse stimuli needs to be confirmed in primate animal experiments and human clinical trials in the future.

## 5. Conclusions

The bench test and computational modeling confirmed the effects of the pulse slope on vestibular afferent recruiting. Consequently, we developed a systematic approach to design stimulation protocols that can enhance the vestibular stimulation performance, particularly in the aging population. More animal experiments need to be carried out in the future to confirm the findings of this study and further explore the potential of the new stimulation protocol. In the future, we aim to offer a stimulation protocol optimization scheme to implanted patients, enabling them to select the most suitable parameters based on their unique brain adaptation schedules and timeframes. This personalized approach seeks to maximize the efficacy and benefits of vestibular stimulation for individual patients.

## Figures and Tables

**Figure 1 bioengineering-10-01436-f001:**
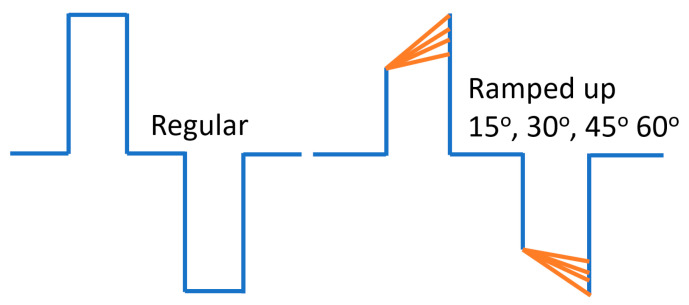
Sketch of a regular pulse and ramp-up pulse edited in customized program in the lab for vestibular afferent. These regular pulse parameters were edited based on the published data. Biphasic stimulus current was delivered via one pair of electrodes immersed in 0.9% NaCl under 240 µA/phase pulses of 50, 120, and 200 µs/phase, with cathodic-to-anodic intrapulse interval set to 10% of the duration of each phase (1).

**Figure 2 bioengineering-10-01436-f002:**
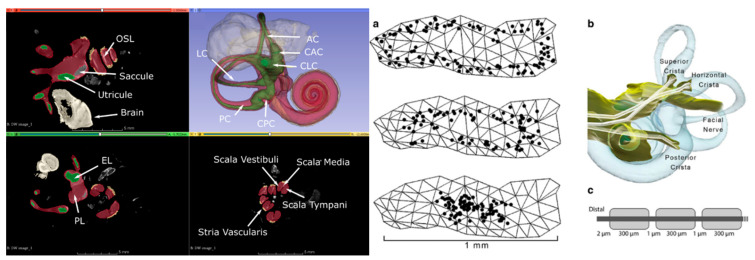
**Left**: Segmented µMRI scan of the rhesus monkey’s inner ear with key structures labeled. The vestibule is shown in green, perilymph in red, saccule in grey, and stria vascularis in yellow. (symbol definitions shown in previous paper) [20]. **Right**: Reprinted from previous paper [18]. (**a**) One hundred model afferent starting positions in peripheral (top image), intermediate (middle), and central (bottom) zones of the posterior semicircular canal crista surface. (**b**) Representative set of fiber trajectories within the three ampullary nerve domains. (**c**) Model geometry of one model vestibular afferent (not to scale).

**Figure 3 bioengineering-10-01436-f003:**
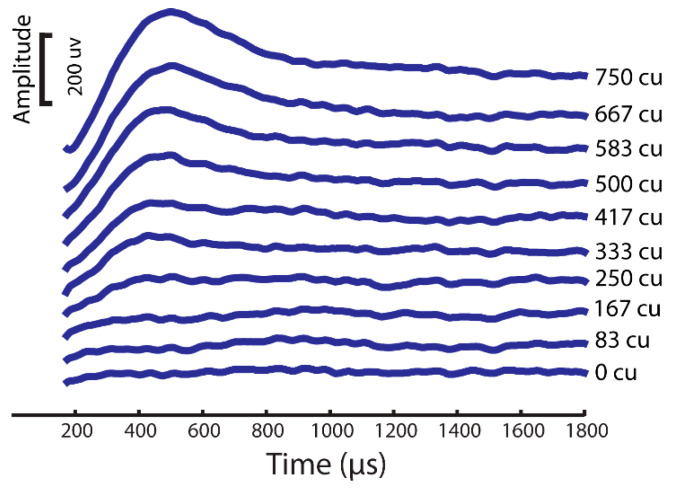
eCAP sample from a rhesus monkey computational modeling simulation with same stimuli as the experimental setup.

**Figure 4 bioengineering-10-01436-f004:**
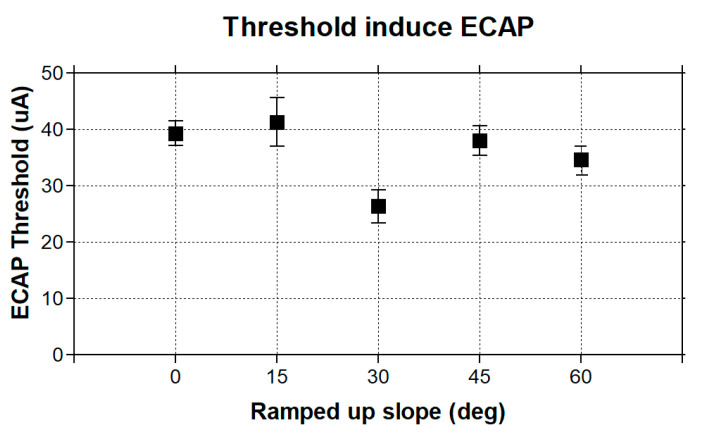
Thresholds of stimuli with four ramped-slope pulses (ramped-up 15, 30, 45, and 60 degrees, as indicated in Figure 1).

**Figure 5 bioengineering-10-01436-f005:**
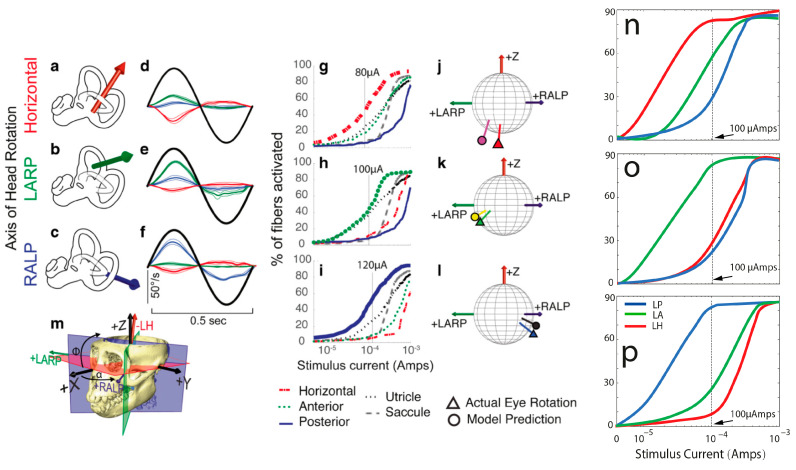
(**a**–**m**) Reprinted from Hedjoudje et al. (2019), and (**n**–**p**) is the simulation result from this study. Comparison of model predictions with empirical data for single set of 3D angular vestibulo-ocular reflex (VOR) responses to 2 Hz sinusoidally pulse-frequency-modulated, symmetric biphasic cathodic-first 200 μS/phase biphasic current pulse stimuli delivered via monopolar electrodes in the left horizontal, anterior, and posterior semicircular canals (SCCs) of M067RhF monkey with its head immobile in darkness and its head-mounted vestibular prosthesis commanded to modulate its input as though the head were moving. (**a**–**c**) Anatomic axis of rotation for each SCC, which approximates the axis of the component of 3D VOR driven by activity of that canal. (**d**–**f**) Means ± SD horizontal (red), left-anterior/right-posterior (LARP, green), and right-anterior/left-posterior (RALP, blue) components of slow-phase nystagmus measured in M067RhF during electrical stimulation via electrodes in the left-horizontal (LH; (**d**)), left-anterior (LA; (**e**)), and left-posterior (LP; (**f**)) SCCs. Black sinusoid indicates equivalent head velocity waveform that would elicit the stimulus rate modulation if a vestibular prosthesis were set to its normal motion-modulated mode of action. The first half cycle of the black sinusoid signifies excitatory stimulation, which typically results in negative horizontal, positive LARP, and positive RALP eye movements when represented by the right-hand-rule coordinate reference frame used throughout this report. (**g**–**i**) Model outputs in the form of fiber recruitment curves showing the relative proportion of model axons activated in each vestibular nerve branch in the first 2 ms following a single 200 μS/phase, cathodic-first, symmetric biphasic pulse of amplitude of 0.01–1 mA, as computed using models individualized to reflect the active electrode’s relative location for each case from which the empiric data in (**d**–**f**) were recorded. The vertical dashed lines indicate the current amplitudes for which the data in (**d**–**f**) were measured. (**j**–**l**) Actual measured (triangle) and model-predicted (circle) mean 3D VOR axes for the cases corresponding to those in (**d**–**i**). Vectors are normalized in amplitude to facilitate comparison of axes of rotation. Although the actual axis of measured eye rotation responses does not always align well with the axis of the ampullary nerve being targeted (likely due to current spread causing spurious stimulation of other nerve branches), the model-predicted axis fits empiric data to within ~20°. (**m**) Three-dimensional reconstruction of a computed tomography scan of F60738RhG (redrawn from Dai et al., 2013) showing best-fit planes for each SCC in the left labyrinth, along with corresponding axes defining the coordinate system used in all other panels. Note that the red axis in (**a**) and (**j**–**m**) is inverted to facilitate display, since excitation of the left-horizontal SCC normally elicits a rightward eye rotation that would be beneath the sphere. In contrast, excitation of the LA and LP SCCs elicits right-hand-rule eye rotations about the +LARP and +RALP axes, respectively. The simulation similar to (**g**–**i**) with ramped slope of 30° is shown in (**n**–**p**).

**Figure 6 bioengineering-10-01436-f006:**
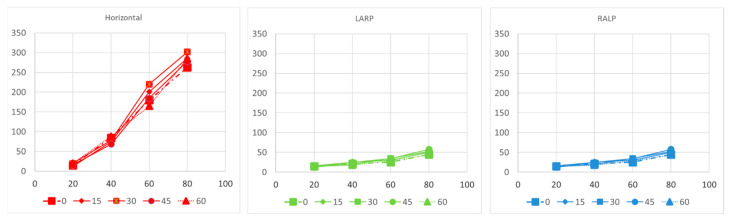
Component-by-component relationship between 3D VOR responses and model predictions for the fraction of afferents activated in each ampullary nerve in modeling simulation. RALP: right-anterior/left-posterior ; LARP: left-anterior/right-posterior.

**Figure 7 bioengineering-10-01436-f007:**
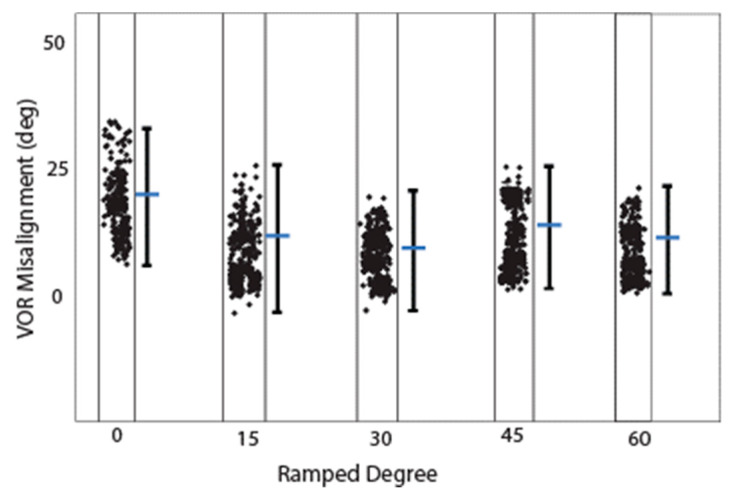
Misalignment calculated from Rhesus monkey computational modeling. Estimated eVOR misalignment (difference between the desired axes of rotations and the corresponding aVOR responses) during simulation with four different ramped pulse stimuli.

## Data Availability

The data presented in this study are available on request from the corresponding author. The data are not publicly available due to the policy of the University of Oklahoma.

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
