# Peer review of "A Virtual Inner Ear Model Selects Ramped Pulse Shapes for Vestibular Afferent Stimulation"

_bioengineering, 2023, doi:10.3390/bioengineering10121436_

Round 1
Reviewer 1 Report
Comments and Suggestions for Authors
Two approaches are described in an attempt to optimize electrical stimulation pulse shapes for vestibular prostheses. The first approach, computational modeling, relies on an FEM model of electrical field spread in the vicinity of the vestibular organ and nerve, based on data from rhesus monkey. ECAP's of vestibular afferent stimulation werde modeled by application of computed extracellular voltages to cable models of sensory axons. The second approach involved current measurements at various locations in a 3D printed inner ear and its comparisons with model simulation results.
Five different wave shapes were tried whereby the 30 degrees ramped-up pulse wave shape turned out to be the most effective one in terms of threshold sensitivity. Whether these results would hold in human subjects will have to be further explored.
A few clarifications in the methods section could possibly enhance the quality of the paper:
L110: the charge per phase was calculated as the pulse width multiplied by the current level amplitude divided by two -> I do not understand this calculation since I assume that charge can be calculated as the integral of the current over time (area of the pulse shape). Thus, the calculation of charge per phase could add the rectangular part of the pulse to the triangular (ramped up) part. Furthermore, it should be specified whether the current level at the beginning or the end of the ramped pulse is used for threshold determination (e.g. in Figure 4). With varying pulse shapes, a better way of comparing threshold sensitivity might be charge per phase.
In describing the variations of pulse shape it could be interesting to show the extracellular voltage waveform generated by the current pulses. Since the electrical impedance of the electrode tissue interface has resistive and capacitive components the voltage waveform for rectangular current pulses will be always ramped, with increased rise time for ramped up current pulses. It is not obvious why there should be an optimum sensitivity for ramps of 30 degrees and some discussion of this aspect might be interesting.
Regarding the bench test I assume that the 3D printed model of the inner ear was generated based on uCT data of a human temporal bone. If so, it should be mentioned and possible differences to a 3D printed model of the inner ear of a rhesus monkey should be discussed. I also assume that the inner ear specimen is fluid filled (0.9 % NaCl?). Can it be assumed that the electrical impedance for the electrode positions within the 3D printed model (material properties compared to bone/tissue?) are similar to impedance values in real temporal bones? Please comment.
Minor comments:
L265: about 3 uA level -> about 30 uA level
Figure 5 and legend: subplots are denoted in the legend and in the text from a to o. There is a 3D image denoted as m which in the legend text (L308) is denoted as M (capital). L311 contains (A) and (J-M) which should be denoted as (a) and (j-m). The three subplots m, n, o are not annotated and eventually should be renamed as n, o, p (if the 3D reconstruction remains denoted as m).
L320: Effect of... -> duplicate subtitle appears in text, please delete
L322: Fig. 2a -> should probably be written as Fig. 6
L330: Fig. 6 -> should probably be written as Fig. 7
L342: bench test and facial nerve response -> how can there be a facial nerve response in a 3D printed inner ear? Please clarify what is meant here.
Author Response
Thank you so much for your important and inspirational comments and suggestions. We revised the manuscript based on each of the comments as listed in below:
L110: the charge per phase was calculated as the pulse width multiplied by the current level amplitude divided by two -> I do not understand this calculation since I assume that charge can be calculated as the integral of the current over time (area of the pulse shape). Thus, the calculation of charge per phase could add the rectangular part of the pulse to the triangular (ramped up) part. Furthermore, it should be specified whether the current level at the beginning or the end of the ramped pulse is used for threshold determination (e.g. in Figure 4). With varying pulse shapes, a better way of comparing threshold sensitivity might be charge per phase.
Thanks for this very important comment. We revised and replaced the original sentence with “the pulse width multiplied by the current level amplitude. For ramped pulses, the charge per phase was calculated as the pulse width multiplied by the rectangular part of the pulse plus the triangular (ramped up) current level amplitude divided by two.”
In describing the variations of pulse shape, it could be interesting to show the extracellular voltage waveform generated by the current pulses. Since the electrical impedance of the electrode tissue interface has resistive and capacitive components the voltage waveform for rectangular current pulses will be always ramped, with increased rise time for ramped up current pulses. It is not obvious why there should be an optimum sensitivity for ramps of 30 degrees and some discussion of this aspect might be interesting.
Thanks for the comments. We agree that it is not obvious why there should be an optimum sensitivity for ramps of 30 degrees. The mechanism behind this finding remains unclear and more physiological measurement needs to be done to explore the possible cause. Currently we are not able to conduct a serial of complex physiological study (e.g. patch-clamp measurements) to investigate the underlying mechanism. We add this discussion in the manuscript.
Regarding the bench test I assume that the 3D printed model of the inner ear was generated based on uCT data of a human temporal bone. If so, it should be mentioned and possible differences to a 3D printed model of the inner ear of a rhesus monkey should be discussed. I also assume that the inner ear specimen is fluid filled (0.9 % NaCl?). Can it be assumed that the electrical impedance for the electrode positions within the 3D printed model (material properties compared to bone/tissue?) are similar to impedance values in real temporal bones? Please comment.
Thank you so much for the comments. We revised the manuscript and clearly described the 3D printed model of inner ear is based on uCT data of a rhesus monkey. Yes, the 3D printed model was filled with 0.9 % NaCl. The impedance of 3D printed ear sensory tissues is similar to real tissues.
Minor comments:
L265: about 3 uA level -> about 30 uA level
Corrected
Figure 5 and legend: subplots are denoted in the legend and in the text from a to o. There is a 3D image denoted as m which in the legend text (L308) is denoted as M (capital). L311 contains (A) and (J-M) which should be denoted as (a) and (j-m). The three subplots m, n, o are not annotated and eventually should be renamed as n, o, p (if the 3D reconstruction remains denoted as m).
Thank you for the suggestions and we corrected the errors and redo the figure.
L320: Effect of... -> duplicate subtitle appears in text, please delete
Corrected
L322: Fig. 2a -> should probably be written as Fig. 6
Corrected
L330: Fig. 6 -> should probably be written as Fig. 7
Corrected
L342: bench test and facial nerve response -> how can there be a facial nerve response in a 3D printed inner ear? Please clarify what is meant here.
Sorry for the confusion. The 3D printed inner ear would not be able to simulate the facial nerve response. The facial response threshold data is referred to comparison to published data.
Reviewer 2 Report
Comments and Suggestions for Authors
Thank you for your interesting paper which is very important to understand the problems with bilateral vestibular deficiency.
For a "normal" physician some passages are difficult to understand from the theoretical point of view. But all in all the take home message has a tremendous impact for the future treatment fo BVD.
Congratulations to your work!
Perhaps you could explain what ffps means?
Author Response
Thank you so much for your encouragement!
Perhaps you could explain what ffps means?
Sorry for the typo, we corrected it in manuscript: firing per second (fps)
Reviewer 3 Report
Comments and Suggestions for Authors the article is interesting and deserves to be publishedAuthor Response
Thank you so much for your kind comment!